# MultiContrievers: Analysis of Dense Retrieval Representations

## Abstract

Dense Retrievers compress source documents into vector representations; the information they encode determines what is available to downstream tasks (e.g., QA, summarisation). Yet there is little analysis of the *information* in retriever representations. We conduct the first analysis comparing the information captured in dense retriever representations as compared to language model representations. To do this analysis, we present **MultiContrievers**, 25 contrastive dense retrievers initialized from the 25 MultiBerts. We use information theoretic probing to analyse how well MultiContrievers encode two example pieces of information: topic and demographic gender (measured as *extractability* of these two concepts), and we correlate this to performance on 14 retrieval datasets covering seven distinct retrieval tasks. We find that: 1) MultiContriever contrastive training increases extractability of both topic and gender, but also has a regularisation effect; MultiContrievers are more similar to each other than MultiBerts, 2) extractability of both topic and gender correlate poorly with benchmark performance, revealing a gap between the effect of the training objective on representations, and desirable qualities for the benchmark 3) MultiContriever representations show strong potential for gender bias, and we do find allocational gender bias in retrieval benchmarks. However, a causal analysis shows that the source of the gender bias is not in the representations, suggesting that despite this potential, current gender bias is coming from either the queries or retrieval corpus, and cannot be corrected by improvements to modelling alone.We additionally find 4) significant variability across random seeds, suggesting that future work should test across a broad spread, which is not currently standard. We release our 25 MultiContrievers (including intermediate checkpoints) and all code to facilitate further analysis.[1]

## 1 Introduction

Dense retrievers (Karpukhin et al., 2020; Izacard et al., 2022; Hofstätter et al., 2021) are the standard retrieval component of retrieval augmented Question Answering (QA) and other retrieval systems, such as fact-checking (Thorne et al., 2018), argumentation, and others. Yet there has been no analysis of the information present in dense retriever representations, nor how it affects retrieval system behaviour. This lack of analytical work is surprising. Retrievers are widespread, and are used for many purposes that require trust: to increase factuality and decrease hallucination (Shuster et al., 2021), and to provide trust and transparency (Lewis et al., 2020) via a source document that has provenance and can be examined. The information the a representation retains from a source document mediates these abilities. Dense retrievers compress inputs documents into N-dimensional representations, so they necessarily emphasise some pieces of information over others. Yet we still do not know what information is de- or over-emphasised, and how this affects retrieval behaviour. For example, a biography of Mary Somerville will contain many details about her: her profession (astronomy and mathematics), her gender (female), her political influence (women's suffrage), her country of origin (Scotland) and others. These will be relevant to different kinds of queries: which ones will be more and less emphasised after compression?

Some of this type of analysis exists for Masked Language Models (MLMs): Lovering et al. (2021) look at linguistic information that is retained (such as subject verb agreement, for the task of gram-

---

[1]URL withheld for anonymity.

maticality judgments) and when lack of this information results in reliance on spurious heuristics, and Orgad et al. (2022) look at gender information and the effect on gender bias in profession classification. But there is no such analysis for retrievers, which optimise a very different objective. We propose to extend this previous analytical work into the retrieval domain. This leads us to the following research questions:

**RQ1**   What information does a retriever represent, and how does this differ from a language model?

**RQ2**   Do differences in this information correlate with performance on retrieval benchmarks?

**RQ3**   Is gender information in retrievers predictive of their gender bias?

To answer these questions, we train 25 **MultiContrievers** initialised from the released MultiBerts (Sellam et al., 2022). We then use information theoretic probing, also known as minimum description length (MDL) probing (Voita & Titov, 2020) to measure the information in **MultiContriever** representations. We evaluate the models on 14 retrieval datasets from the BEIR benchmark Thakur et al. (2021). We find that:

(**RQ1**) For both MultiBerts and MultiContrievers, gender is more extractable than topic, but there are noticeable differences in the models. MultiContrievers have *both* more extractable topic and more extractable gender, but a lower *ratio* between the two. MultiContrievers have overall richer representations for topic and gender, but they still have potential to rely on gender heuristics (which are a source of gender bias). MultiContrievers also have a smaller range of extractability between the 25 seeds, showing a regularisation effect of their training.

(**RQ2**) Despite the increase in extractability of both gender and topic, neither correlates with retrieval performance on benchmark datasets. This highlights the gap between features encouraged by the retriever training process, and those tested by the benchmarks, and our findings suggest it is a limitation of the benchmark. When we subsample a set of questions that require gender information to answer them correctly, we do see a correlation between gender extractability and performance. This indicates that gender information is used by the model, but that most questions in the benchmark can be answered without it. Retrieval benchmarks may underspecify some desirable characteristics of a good model.

(**RQ3**), despite the evidence that extractability of gender information is helpful to the model, it is *not* the cause of allocational gender bias in the Natural Questions (NQ) dataset. We find that when we do a causal analysis by removing gender from MultiContriever representations, gender bias persists, indicating that the source of bias is in the queries or corpus.

Our contributions are: **1)** the first information theoretic analysis of dense retrievers for performance and for fairness, **2)** the first causal analysis of social bias in dense retrievers, to identify the source of the bias, **3)** a broad analysis of variability in performance and fairness across random retriever seeds, **4)** a suite of 25 **MultiContrievers** for use in future work, as well as a small gender annotated subset of Natural Questions, and all training and evaluation code.

## 2   BACKGROUND AND RELATED WORK

### 2.1   WHAT IS A RETRIEVER?

Retrievers take an input query and return relevance scores for documents from a corpus. We use the common dense retrieval approach where documents $D$ and queries $Q$ are encoded separately by the same model $f_\theta$, and relevance is given by the dot product between them. For a given query $q_i$ and document $d_i$, the relevance score $s$ is:

$$s(d_i, q_i) = f_\theta(q_i) \cdot f_\theta(d_i) \tag{1}$$

Training $f_\theta$ is a challenge. Language models like BERT (Devlin et al., 2019), are not good retrievers out-of-the-box, but retrieval training resources are limited and expensive to create, since they involve matching candidate documents to a query from a corpus of potentially millions. So retrievers are either trained on one of the few corpora available, such as Natural Questions (NQ) (Kwiatkowski et al., 2019) or MS MARCO (Campos et al., 2016) as supervision (Hofstätter et al.,

2021; Karpukhin et al., 2020), or on a self-supervised proxy for the retrieval task (Izacard et al., 2022). Both approaches results in a domain shift between training and later inference, making retrieval a *generalisation task*. This motivates our analysis, as Lovering et al. (2021)'s work showed that information theoretic probing was predictive of where a model would generalise and where it would rely on simple heuristics and dataset artifacts.

In this work, we focus on the self-supervised Contriever (Izacard et al., 2022), initiaised from a BERT model and then fine-tuned with a contrastive objective. For this objective, all documents in a large corpus are broken into chunks, where chunks from the same document are positive pairs and chunks from different documents are negative pairs. This is a loose proxy for "relevance" in the retrieval sense, so we are interested in what information this objective encourages contriever to emphasise, what to retain, and what to lose, and what this means for the eventual retrieval task.

### 2.2 How do we find out what information is in a retriever?

The most common analysis of what information is in a model representation is via **probing**, also called 'diagnostic classifiers' Belinkov & Glass (2019). Let $D = \{(d_i, y_i), ..., d_n, y_n)\}$ be a dataset, where $d$ is a document (e.g. a chunk of a Wikipedia biography about Mary Somerville) and $y$ is a label from a set of $k$ discrete labels $y_i \in Y$, $Y = \{1, ...k\}$ for some information in that document (e.g. *mathematics, astronomy* if probing for topic).

In a probing task, we want to measure how well $f_\theta(d_i)$ encodes $y_i$, for all $d_{1:n}$, $y_{1:n}$. We use Minimum Description Length (MDL) probing (Voita & Titov, 2020), or information theoretic probing, in our experiments. This measures **extractability** of $Y$ via compression of information $y_{1:n}$ from $f_\theta(d_{i:n})$ via the ratio of uniform codelength to online codelength.

$$Compression = \frac{L_{uniform}}{L_{online}} \tag{2}$$

where $L_{uniform}(y_{1:n}|f_\theta(d_{i:n}) = n \log_2 k$ and $L_{online}$ is calculated by training the probe on increasing subsets of the dataset, and thus measures quality of the probe relative to the number of training examples. Better performance with less examples will result in a shorter online codelength, and a higher compression, showing that $Y$ is more extractable from $f_\theta(d_{i:n})$.

In this work, we probe for binary *gender*, where $Y = \{m, f\}$ and *topic*, where $Y = \{lawyer, doctor, ...\}$

### 2.3 Why does it matter?

**Extractability**, as measured by MDL probing, is predictive of *shortcutting* (Lovering et al., 2021); when a model relies on a heuristic feature to solve a task, which has sufficient correlation with the actual task to have high accuracy on the training set, but is not the true task (Geirhos et al., 2020). Shortcutting causes failure to generalise; a heuristic that worked on the training set due to a spurious correlation will not work after a distributional shift (Gururangan et al., 2018). This would severely affect retriever performance, which depends on generalisation.

Shortcutting is also often the cause of social biases. Extractability of gender information in language models is predictive of gender bias in coreference resolution and biography classification (Orgad et al., 2022). So when some information, such as gender, is more extractable than other information, such as anaphora resolution, the model is risk of using gender as a heuristic, if the data supports this usage. And thus of both failing to generalise and of propagating biases. For instance, for the case of Mary Somerville, if gender is easier for a model to extract than profession, then a model might have actually learnt to identify mathematicians via *male*, instead of via *maths* (the true relationship), since it is both easier to learn and the error penalty on that is small, as there are not many female mathematicians.

## 3 Methodology

Our research questions require that we analyse the relationship between information in different model representations, and their performance and fairness. This requires at minimum a model, a probing dataset (with labels for information we want to probe for), and a performance dataset. We

need some performance datasets to have demographic labels so that we can calculate performance difference across demographics, also called allocational fairness.

To answer **RQ1** (what information is in retriever representations and how does it differ from LMs) we train 25 MultiContrievers with 25 random seeds. We run information theoretic probing on them with two datasets with gender and topic labels, and repeat this for the 25 MultiBerts that they were trained on. We also compare this to the results of some other supervised retrieval models. For **RQ2** we evaluate all models on the 14 BEIR benchmark datasets and correlate to the values from RQ1. For **RQ3** we subsample Natural Questions to queries about entities, and annotate those that have explicit gender as male/female. E.g. an entity query is *Who was the first prime minister of Finland?*, a female query is *Who was the first female prime minister of Finland?* and a male query is *Who was the first male prime minister of Finland?*. We measure performance separately on the male and female query sets, as well as on the general entity sets as a control.

## 3.1 MODELS

For the majority of our experiments, we compare our 25 MultiContriever models to the 25 Multi-Berts models (Sellam et al., 2022). We access the MultiBerts via huggingface[2] and train the con-trievers via modifying the repository released in Izacard et al. (2022). We use the same contrastive training data as Izacard et al. (2022), to maximise comparability with previous results. This comprises a 50/50 mix of Wikipedia and CCNet from 2019. As a result, five of the fourteen performance datasets involve temporal generalisation, since they postdate both the MultiContriever and the MultiBert training data. This most obviously affects the TREC-COVID dataset (QA), though also four additional datasets: Touché-2020 (argumentation), SCIDOCS (citation prediction), and Climate-FEVER and Scifact (fact-checking). As in Izacard et al. (2022), we train for 500,000 steps, saving intermediate checkpoints, sometimes (though rarely) selecting an earlier checkpoint if the model appeared to converge earlier. Further details on contriever training and infrastructure are in Appendix A.

We train 25 random seeds as both generalisation and bias vary greatly by random seed initialisation (McCoy et al., 2020). MultiContrievers have no new parameters, so the random seed affects only their data shuffle. The MultiBerts each have a different random seed for both weight initialisation and data shuffle.

## 3.2 PROBING DATASETS

We use two datasets for probing, to verify that results are not dataset specific or due to any dataset artifacts. First the BiasinBios dataset (De-Arteaga et al., 2019), which contains biographies scraped from the web annotated with labels of the subject's binary gender (male, female) and biography topic (lawyer, journalist, etc). We also use the Wikipedia dataset from md_gender (Dinan et al., 2020), which contains Wikipedia pages about people, annotated with binary gender labels.[3] For gender labels, BiasinBios is close to balanced, with 55% male and 45% female labels, but Wikipedia is very imbalanced, with 85% male and 15% female. For topic labels, BiasinBios has a long-tail zipfian distribution over 28 professions, with professor and physician together as a third of examples and rapper and personal trainer as 0.7%. Examples from both datasets can be found in Appendix B.

To verify the quality of each dataset's labels, we manually annotated 20 random samples and compared to gold labels. BiasinBios agreement with our labels was 100%, and Wikipedia's was 88%.[4] We focus on the higher quality BiasinBios dataset for most of our graphs and analysis, though we replicate all experiments on Wikipedia.

---

[2]e.g. `https://huggingface.co/google/MultiBerts-seed_[SEED]`

[3]This dataset does contain non-binary labels, but there are few (0.003%, or 1̄80 examples out of 6 million). Uniform codelength ($dataset\_size * log2(num\_classes)$) is important to information theoretic probing, so an additional class with very few examples can significantly affect results. This dataset was also noisier, making small subsets of data even less trustworthy.

[4]We investigated other md_gender datasets in the hope of replicating these results on a different domain such as dialogue (e.g. Wizard of Wikipedia), but found the labels to be of insufficiently high agreement to use.

### 3.3 EVALUATION DATASETS AND METRICS

We evaluate on fourteen publicly available datasets in the BEIR benchmark. BEIR covers retrieval for seven different tasks (fact-checking, citation prediction, duplicate question retrieval, argument retrieval, question answering, bio-medical information retrieval, and entity retrieval).[5]. We initially analysed all standard metrics used in BEIR and TREC (e.g. NDCG, Recall, MAP, MRR, @10 and @100). We observed similar trends across all metrics somewhat to our surprise, since many retrieval papers focus on the superiority of a particular metric (Wang et al., 2013). We thus predominantly report NDCG@10, as it is standard on the BEIR benchmark benchmark (Thakur et al., 2021).

For allocational fairness evaluation, we create a subset of Natural Questions (NQ) about entities, annotated with male, female, and neutral (no gender). We subsample Natural Questions to entity queries by filtering for queries containing any of `who, whose, whom, person, name`. We similarly filter this set into gendered entity queries by using a modified subset of gender terms from Bolukbasi et al. (2016). This automatic process is low precision/high recall[6] so we manually filter these results by annotating with two criteria: gender of the subject (male, female, or neutral/none[7]), and a binary tag with whether the query actually *constrains* the gender of the answer. This second annotation is somewhat subtle. For example, in our dataset there is the query `Who was the actress that played Bee`, which contains a gendered word (actress) but it is not necessary to answer the question; all actors that played Bee are female, and the question could be as easily answered in the form `Who played Bee?`. Whereas in another example query, `Who plays the sister in Home Alone 3?` the query does constrain the gender of the answer. We annotated 816 queries with both of these attributes, of which 51% have a gender constraint, with a gender breakdown of 59% female and 41% male.

We measure allocational fairness by the difference between the female and male query performance. We use the neutral/no gender entity queries as a control to make sure the system performs normally on this type of query.

## 4 RESULTS

Below we address the three research questions: how does extractability change (RQ1), do the changes correlate with performance (RQ2), and is this predictive of allocational bias (RQ3). We also analyse the overall performance and quality of the MultiContrievers, as this is the first study that includes variability over a large number of retriever initialisations, with some surprising results from this alone.

### 4.1 MULTICONTRIEVERS OVERVIEW

We analysed the distribution of performance by dataset for 24 seeds, to ensure that our MultiContrievers have competitive performance, which strengthens both our analysis and their utility to future researchers.[8] Figure 1 shows this data, broken out by dataset, with a dashed line at previous reference performance (Izacard et al., 2022). Table 1 shows the best and worst individual seed per dataset.

A few things are notable: first, **there is a large range of benchmark performance across seeds with for identical contrastive losses.** During training, MultiContrievers converge to the same accuracy (see Appendix A) and (usually) have the same aggregated BEIR performance reported in Izacard et al. (2022). However, the range of scores per dataset is often quite large, and for some datasets the original Contriever is in the tail of the distribution: e.g in Climate-Fever (row 1 col-

---

[5]The BEIR benchmark itself contains two additional tasks, tweet retrieval, and news retrieval, but these datasets are not publicly available.

[6]It captures queries with gendered terms in prepositional phrases, (`Who starred in O Brother Where Art Thou?`) which are common false positives in QA datasets, as they are not about brothers.

[7]In cases where the gender term was actually in a title or other prepositional phrase as in the example

[8]Seed 13 (ominously) is excluded from our analysis because of extreme outlier behaviour, which was not reported in Sellam et al. (2022). We investigated this behaviour, and it is fascinating, but orthogonal to this work, so we have excluded the seed from all analysis. Our investigation can be found in Appendix D and should be of interest to researchers investigating properties of good representations (e.g. anisotropy) and of random initialisations.

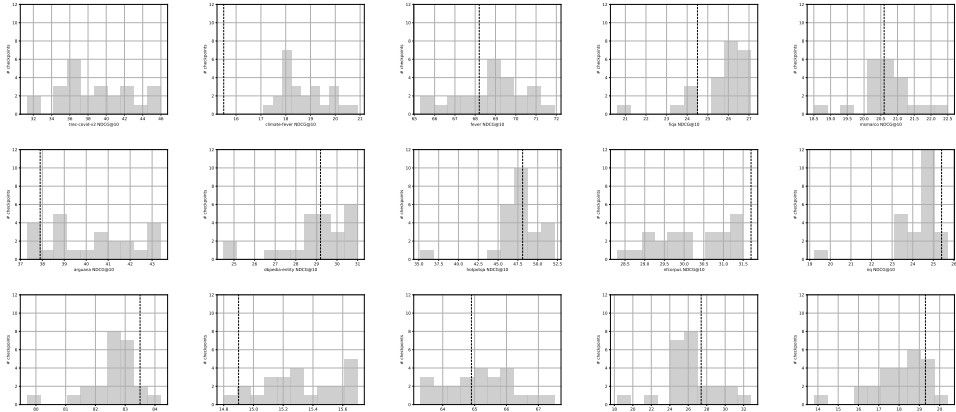

Figure 1: Distribution of performance (NDCG@10) for the 24 MultiContrievers, per BEIR dataset. Dashed line indicates reference performance from previous work. While for some datasets the reference performance sits at or near the mean of the MultiContriever distribution, for some the reference performance is an outlier.

| | arguana | climate-fever | fiqa | nf-corpus | scidocs | scifact | trec-covid | webis-touche | dbpedia-entity | fever | hotpotqa | msmarco | nq | quora | all |
|---|---|---|---|---|---|---|---|---|---|---|---|---|---|---|---|
| best_seed | 16 | 5 | 1 | 6 | 0 | 5 | 10 | 8 | 17 | 0 | 8 | 23 | 4 | 24 | 24 |
| worst_seed | 8 | 10 | 4 | 14 | 5 | 10 | 19 | 18 | 10 | 18 | 10 | 10 | 10 | 20 | 10 |
| delta | 6.1 | 3.8 | 6.3 | 3.2 | 0.9 | 4.2 | 14.5 | 6.6 | 6.5 | 6.6 | 16.9 | 4 | 6.5 | 4.5 | 3.9 |

Table 1: Best and worst performing seeds per BEIR dataset, with delta in NDCG@10

umn 2) it performs *much* worse than all 24 models. It is worse than almost all models for Fiqa and Arguana, for Fiqa 19 models are up to 2.5 points better, for Arguana 20 models are up to 6.3 points better. Nothing changed between the different MultiContrievers except the random seed for MultiBert initialisation, and the random seed for the data shuffle for contrastive fine-tuning.[9]

Second, **the difference in performance across random seeds can exceed the difference in performance from adding supervision (over unsupervised learning only)**; we see this effect for half the datasets in BEIR. The higher performing seeds surpass the performance on *all* supervised models from Thakur et al. (2021)[10] on three datasets (Fever, Scifact, and Scidocs) and surpass all but one model (TAS-B) on Climate-fever. These datasets are the fact-checking and citation prediction datasets in the benchmark, suggesting that even under mild task shifts from supervision data (which is always QA), random initialisation can have a greater effect than supervision. This effect exists across diverse non-QA tasks; for four additional datasets the best random seeds are better than all but one supervised model: this is true for Arguana and Touché (argumentation), HotpotQA (multihop QA), and Quora (duplicate question retrieval).

Third, Table 1 shows that **the best and worst model across the BEIR benchmark datasets is not consistent**; not only is the range large across seeds but the ranking of each seed is very variable. The best model on average, seed 24, is top-ranked on only *one* dataset, and the second-best average model, seed 2, is best on *no* individual datasets. The best or worst model on any given dataset is almost always the best or worst on *only* that dataset and none of the other 14. Sometimes, the best model on one dataset is worst on another, e.g. seed 4 is best on NQ and worst on FiQA, seed 5 is best on Scifact and worst on Scidocs.[11] Even seed 10, which is the only model that is worst on more than 2 datasets (it is worst on 6) is still best on TREC-Covid.[12] This is the most clear case of generalisation, as these models are trained on only pre-Covid data. Our results show that there is no single best retriever, which both supports the motivation of the BEIR benchmark (to give a more

---

[9]There are a few small differences between the original released BERT, which Contriever was trained on, and the MultiBerts, which we trained on, detailed in Sellam et al. (2022).

[10]The BEIR benchmark reports performance on all datasets for four dense retrieval systems—DPR(Karpukhin et al., 2020), ANCE (Xiong et al., 2021), TAS-B (Hofstätter et al., 2021), and GENQ (their own system)—which all use supervision of some kind. DPR uses NQ and Trivia QA, as well as two others, ANCE, GENQ, and TAS-B all use MSMARCO. The original contriever underperformed these other models until supervision was added.

[11]This best-worst flip exists for seeds 8, 18, and 23 also.

[12]Analysis on seed 13 revealed that seed 10 was also unusual, that analysis can also be found in D.

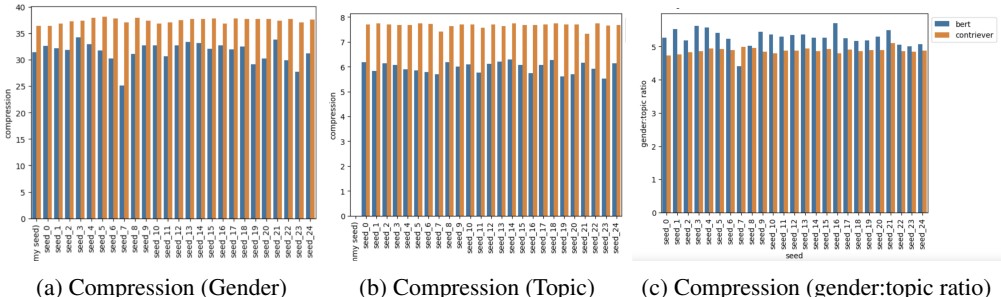

Figure 2: Comparison of Bert and Contriever compression for gender and topic on the BiasinBios dataset, over all seeds. Contriever has more uniform compression across seeds, and a lower ratio of gender:topic.

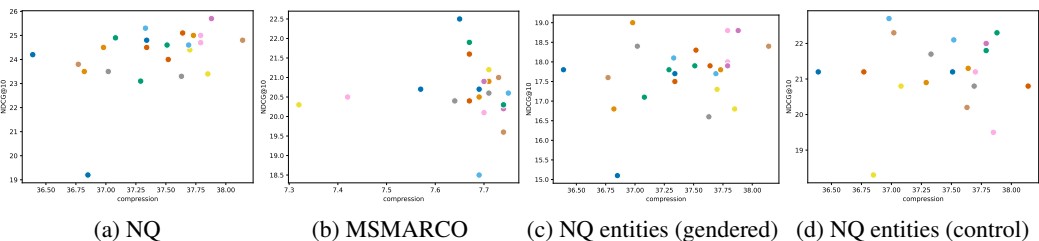

Figure 3: Scatterplots of the correlation between x-axis compression (ratio of uniform to online codelength) and y-axis performance (NDCG@10), for different datasets (NQ, MSMARCO) at left and entity subsets of NQ at right. Different seed colour is held constant across graphs.

well rounded view on retriever performance via a combination of diverse datasets) and shows the need for more analysis into this phenomena .

## 4.2 RQ1: INFORMATION EXTRACTABILITY

Figure 2 shows extractability of gender 2a and topic 2b on the BiasinBios dataset, for MultiContriever and MultiBert models. 2c shows the gender:topic ratio. These graphs show a few differences both between the two types of information and the two models. Both gender and topic are more extractable in MultiContrievers than MultiBerts. Gender compression ranges for MultiContrievers are 4-12 points higher, or a 9-47% increase (depending on seed initialisation), than the corresponding MultiBerts. Topic compression ranges are 1.7-2.12 points higher for MultiContrievers; as the overall compression is much lower this is a 19-38% increase over MultiBerts. Figure 2 also shows a regularisation effect; MultiBerts have a large range of compression across random seeds, whereas most MultiContrievers have similar values.

Figure 2c shows that though MultiContrievers have higher extractability for gender and topic, the ratio between them decreases; the contrastive training encourages both topic and gender, but increases topic at a greater rate. So while MultiContrievers do represent gender far more strongly than topic, this effect is lessened vs. MultiBerts, which means they should be slightly less likely to shortcut based on gender (Lovering et al., 2021).

## 4.3 RQ2: DOES INFORMATION EXTRACTABILITY CORRELATE WITH PERFORMANCE?

Figures 3 3a and 3b show correlation between gender compression and performance (NDCG@10) on NQ and between topic compression and performance on MSMARCO. NQ and MSMARCO are the most widely used of the BEIR benchmark datasets, and are the datasets that we hypothesised were most likely to correlate. Both datasets are search engine queries (from Google and Bing, respectively) and thus contain queries that require topic information (*what is cabaret music?*, MS-MARCO) and queries that require gender information (*who is the first foreign born first lady?*, NQ). However, as the dispersed points on the scatterplots in Figure 3a and 3b show, neither piece of information correlates to performance on either dataset. NQ and MSMARCO are representative; we include plots for all datasets in Appendix C. We tested on the average over all datasets, on each

dataset individually, and on each retrieval metric, and found only a few isolated cases of correlations (discussed also in C).

This result was somewhat surprising; lack of correlation between extractability and performance points to a mismatch between the self-supervised contrastive training objective that is a proxy for retrieval, and retrieval benchmarks. The contrastive training both regularises and increases extractability of gender and topic, but perhaps it is relevant for *only* that objective, and not for the retrieval benchmark. Alternatively, it is possible that this information is important, but only up to some threshold that MultiContriever models exceed. Finally, it's possible that this information doesn't matter for most queries in these datasets, and so there is some correlation but it is lost, as these datasets are extremely large. This is somewhat supported by the exception cases with correlations being smaller, more curated datasets (C), and so we investigate this as the most tractable to implement.

 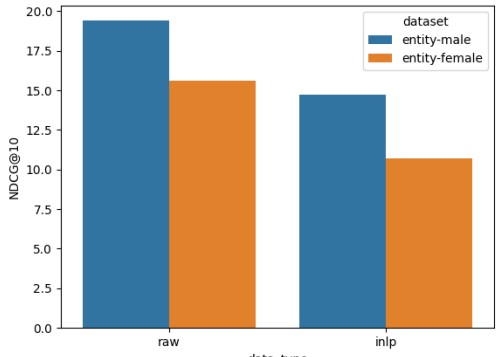

(a) Performance on the no-gender-constraint NQ entity subset vs. the gendered NQ entity subset that. Raw representations (blue) vs. INLP representations (orange) where gender has been removed. INLP performance degrades on only gender constrained queries, showing that gender information is used in those queries but not in the control.

(b) Causal analysis of allocational gender bias in the NQ gendered entity queries, measured as difference in performance between male (blue) and female (orange) entity queries. When gender is removed (as in INLP representations), the gap in performance remains, showing that bias is not due attributable to gender in the representations.

Figures 3c and 3d show correlation on our two subsets of NQ: gendered queries and non-gendered queries (§3.3). The gendered entity queries show a correlation, and the non-gendered control shows none. If we isolate to a topical dataaset, extractability is predictive of performance, it just is not over the whole diversity of a large dataset.

We strengthen this analysis, testing whether the gender information is necessary, rather than simply correlated. We use Iterative Nullspace Projection (INLP) (Ravfogel et al., 2020) to remove gender information from MultiContriever representations; INLP learns a projection matrix $W$ onto the nullspace of a gender classifier, which we apply before computing relevance scores between corpus and query. So with INLP, Equation 1 becomes:

$$s(d_i, q_i) = \mathbf{W}f_\theta(q_i) \cdot \mathbf{W}f_\theta(d_i) \tag{3}$$

Then we calculate performance of retrieval with these genderless representations. If there is no drop in performance on either of the sets of queries, then gender information not necessary for it, if there is a drop, then it was necessary. If there is a drop in performance on *both* gender and control queries, then the threshold explanation may be true, but the representation was sufficiently degraded by the removal of gender that the experiment is difficult to interpret.

When we perform INLP, the gender information drops to 1.4 (nearly none, as 1 is no compression over uniform, see Eq 2). Figure 4a show that performance on non-gendered entity queries is unaffected, but performance on gendered entity questions drops significantly (5 points). From these two experiments we conclude that the increased information extractability from the contrastive training *was* useful for answering specific questions that require that information. But most queries in the available benchmarks simply don't require that information to answer them.

### 4.4 RQ3: IS GENDER EXTRACTABILITY PREDICTIVE OF ALLOCATIONAL GENDER BIAS?

Orgad et al. (2022) found gender extractability in representations to be predictive of allocational gender bias for classification tasks; when gender information was reduced or removed, bias also reduced.[13] We found in RQ2 that information is *used* so now we ask: is it predictive, as it was in (Orgad et al., 2022)? Figure 4b shows that, at least for our dataset, it is not. It shoes allocational bias between the female and male queries, and the bias that remains *after* we remove gender via INLP. *All* performance drops, as we saw in RQ2, but by equivalent amounts for female and male entities. These results are surprising, and suggest that allocational gender bias in this case does not come from the representations, but instead from the retrieval corpus or the queries, or from a combination. The corpus could have lower quality or less informative articles about female entities, queries about women could be structurally harder in some way.

## 5 DISCUSSION, FUTURE WORK, CONCLUSION

We trained a suite of 25 **MultiContrievers**, analysed their performance on the BEIR benchmark, probed them for gender and topic information, and removed gender information from their representations to analyse allocational gender bias.

Our experiments showed that performance itself is extremely variable by random seed initialisation, as is the ranking of different random seeds per dataset, despite all models having equivalent contrastive losses during the training. Best seed performances often exceed the performance of more complex dense retrievers that use explicit supervision. This suggests that future analysis of retriever loss basins to look for differing generalisation strategies would be valuable (Juneja et al., 2023). Our results show that this work may be more valuable than developing new models, as random seed initialisation can lead to greater performance improvements. Our work also highlights the usefulness of labelled dataset not just for supervision but for analysis. Future work could create these datasets and then probe for additional targeted information. It could also analyse demographic biases beyond binary gender, such as race or sexual orientation, or even against different demographic dialects in argumentation datasets.

We showed that gender and topic extractability is not predictive of performance except in subsets of queries that clearly require gender information, despite a strong increase in both during Multi-Contriever training. We showed that though both gender and topic increase, the ratio of gender to topic decreases. However, since it remains large, these models are likely to shortcut based on gender (Lovering et al., 2021). Despite this finding, the gender bias we find is not a product of the representations, as it persists when gender is removed.[14] More research should be done on where is best in a pipeline to correct bias, and how various parts interact. This work also shows the utility of information removal (INLP and others) for causality and interpretability, rather than just debiasing. Future research could construct tests for shortcutting to increase the scope of these preliminary results.

Finally, we have analysed only the retriever component of a retrieval system. In any eventual downstream task of retrieval augmented generation, the retrieval representation will have to compete with language model priors, such that the eventual generation is a composition between text that was unconditionally probable and text that is attested by the retrieved data. Future work should investigate the role of information extractability in the full system, and how this bears on vital questions like hallucination in retrieval augmented generation. We have done the first information theoretic analysis of retrieval systems, and the first causal analysis of the reasons for allocational gender bias in retrievers, and we have raised many new questions for the research community. We release our code and resources for the community to expand and continue this line of enquiry. This is particularly important in the current generative NLP landscape, which is increasingly reliant on retrievers and where understanding of models lags so far behind development.

---

[13]Orgad et al. (2022) actually use a lexical method of removing gender, but we chose to use INLP as a more elegant and extensible solution. We were able to replicate their paper using INLP, showing equivalence.

[14]These two results are not necessarily conflicting – allocational bias *can* be caused by shortcutting behaviour, as is the case in Zhao et al. (2018) where allocational bias is the result of stereotyping heuristics, but can also exist for other reasons, such as differences in training data quantity or quality, as in Tatman (2017). So allocational bias in NQ is the result of another cause, but this may be a property of NQ rather than of allocational bias in retrievers generally.

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

## A  CONTRIEVER TRAINING

Each MultiContriever model was initialised from a MultiBert checkpoint for each of the 25 seeds from 0 - 24, accessed at `https://huggingface.co/google/multiberts-seed_X` where `X` is an integer from 0 - 24. NB: MultiBerts released many checkpoints to enable study of training dynamics, we use only the final complete checkpoint.

Hyperparameters and training regime is exactly matched to the original Contriever work of (Izacard et al., 2022). Hyperparams can be found in Table 2. Data used was identical to in (Izacard et al., 2022) (from 2019) and was a 50/50 CCNet Wikipedia split.

Each MultiContriever was trained across 4 nodes with 8 GPUs per node (32 GPUs total) for on average 2.5 days. Each MultiContriever was trained for the full 500,000 steps, and checkpointed often; but in all cases the best performing checkpoint was the final one, save seed 13, which was anomalous in many other ways (see D).

All MultiContrievers have similar loss and accuracy curves, with seeds 12 and 13 excerpted in Figure 5. All models steeply increase accuracy/decrease loss within 10,000 steps, and then asymptotically approach 69% accuracy by 50,000 steps.

| | |
|---|---|
| sampling coefficient | 0 |
| pooling | average |
| augmentation | delete |
| probability_augmentation | 0.1 |
| momentum | 0.9995 |
| temperature | 0.05 |
| queue_size | 131072 |
| chunk_length | 256 |
| warmup_steps | 20000 |
| total_steps | 500000 |
| learning_rate | 0.00005 |
| scheduler | linear |
| optimizer | adamw |
| batch_size (per gpu) | 64 |

Table 2: Hyperparameters used for training all MultiContrievers.

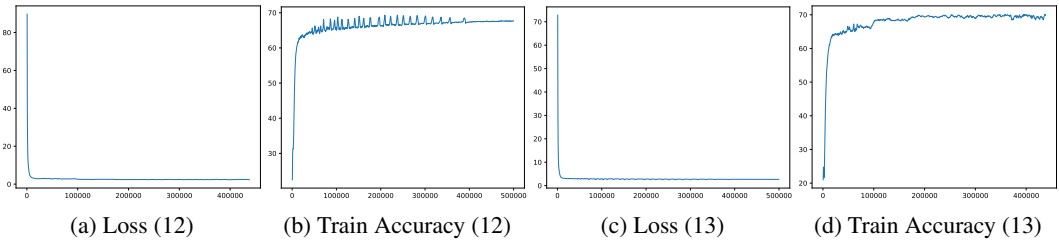

(a) Loss (12)    (b) Train Accuracy (12)    (c) Loss (13)    (d) Train Accuracy (13)

Figure 5: Loss and accuracy for seeds 12 and 13, steps on x-axis and loss or accuracy on y-axis.

## B  PROBING DATASETS

We rely on two datasets. The first is BiasinBios (De-Arteaga et al., 2019), which is a dataset of web biographies labelled with binary gender, and biography profession. We use De-Arteaga et al. (2019)'s train/dev/test splits of 65:10:25, yielding 255,710 train 39,369 dev, and 98,344 test data-points. Second is the Wikipedia slice of the md_gender dataset (Dinan et al., 2020). This has only labels for gender, which we restrict to be binary since non-binary gender is so small and would adversely affect this analysis. We filter out texts below 10 words (words, not tokens) leaving a dataset of size 10,681,700, split 65:10:25 into 6,943,105 train, 934,649 dev, 2,803,946 test. For practical reasons, we shard it to 9 shards (650,000 train examples each) and then check the results on each shard. All shards behaved consistently. As noted in the text, BiasinBios is nearly balanced with regard to gender labels, but Wikipedia is severely imbalanced.

For both datasets, we use the train set for probing, and the test set for measuring accuracy on the final probe. We investigated using other datasets, but none were of sufficient quality that they were usable. We tested usability very simply: each of the authors labelled a different random sample of 20 examples by hand, and we measured accuracy of dataset labels against our labels, and only took datasets with over 80% accuracy, since our probing task is sensitive to errors in labelling. No other subsets of md_gender nor external datasets that we surveyed passed this bar. We didn't multiply annotate as we found no examples to be at ambiguous.

## C  FULL SET OF RESULTS FOR CORRELATION BETWEEN EXTRACTABILITY AND PERFORMANCE

Full set of correlations between gender compression and performance in Figure 6 and between profession compression and performance in Figure 7. The latter (profession correlation) have misleading regression lines as only three of 24 models had large differences in compression, such that the line is based off insufficient datapoints. It is included for completeness but left out of analysis for that reason. Gender compression numbers (Figure 6) are distributed more evenly. There are four statistically significant correlations (referred to as by row 1-4, and column a-d, such that the upper left cell is 1a and the lower right cell is 4d). Arguana (1a), Scifact (2b), Webis-Touche (3a), and NQ (4b). All have middling correlation coefficients: Arguana -0.41, Scifact 0.41, Webis-Touche 0.31, NQ 0.42. There is also little in common between these datasets, Arguana and Webis-Touche are argumentation, Scifact is fact-checking, and NQ is google-search style questions. As this leaves most datasets with no correlations, we consider the correlation overall to be weak. We do note that the temporal generalisation datasets are overrepresented in this set (Webis-Touche and Scifact), but leave an investigation of that for future work.

Arguana in particular is unique in having a significant *negative* correlation. We have no answers as to why this might be. It may be a fluke due to peculiarities of this dataset: the dataset is small (less thank 2k datapoints), and is not structured in the same way with query (input) and passage (retrieved) but instead uses a full document passage as the query. It is unclear why this might cause a deterioration in performance from better gender or profession encoding (as we observe the same in profession compression). The Arguana task should match the unsupervised training much more closely since they both are matching the relevance of to document chunks. We leave an investigation into the peculiarities of that dataset also to future work.

## D  SEED 13

MultiContrievers were trained with seeds 0-24 based on respective MultiBerts 0-24. Seed 13 was excluded from all analysis as it displayed repeatedly anomalous behaviour. During the course of contriever training it appeared indistinguishable from other seeds, loss curves looked normal, there were no signs of overfitting. Performance converged to the same level as other MultiContrievers. However, when applied to the datasets of the BEIR benchmark it did not perform at all, with NDCG of between 2 and 20 on each dataset. We retrained once to replicate the behaviour, and then twice more with different seeds for data shuffle, with identical results. We thus exclude it from all analysis. To aid in future investigations we include our initial analysis of seed 13 irregularities here. We follow the method of analysis of representation spaces from Ethayarajh (2019). We measure the L2 norm

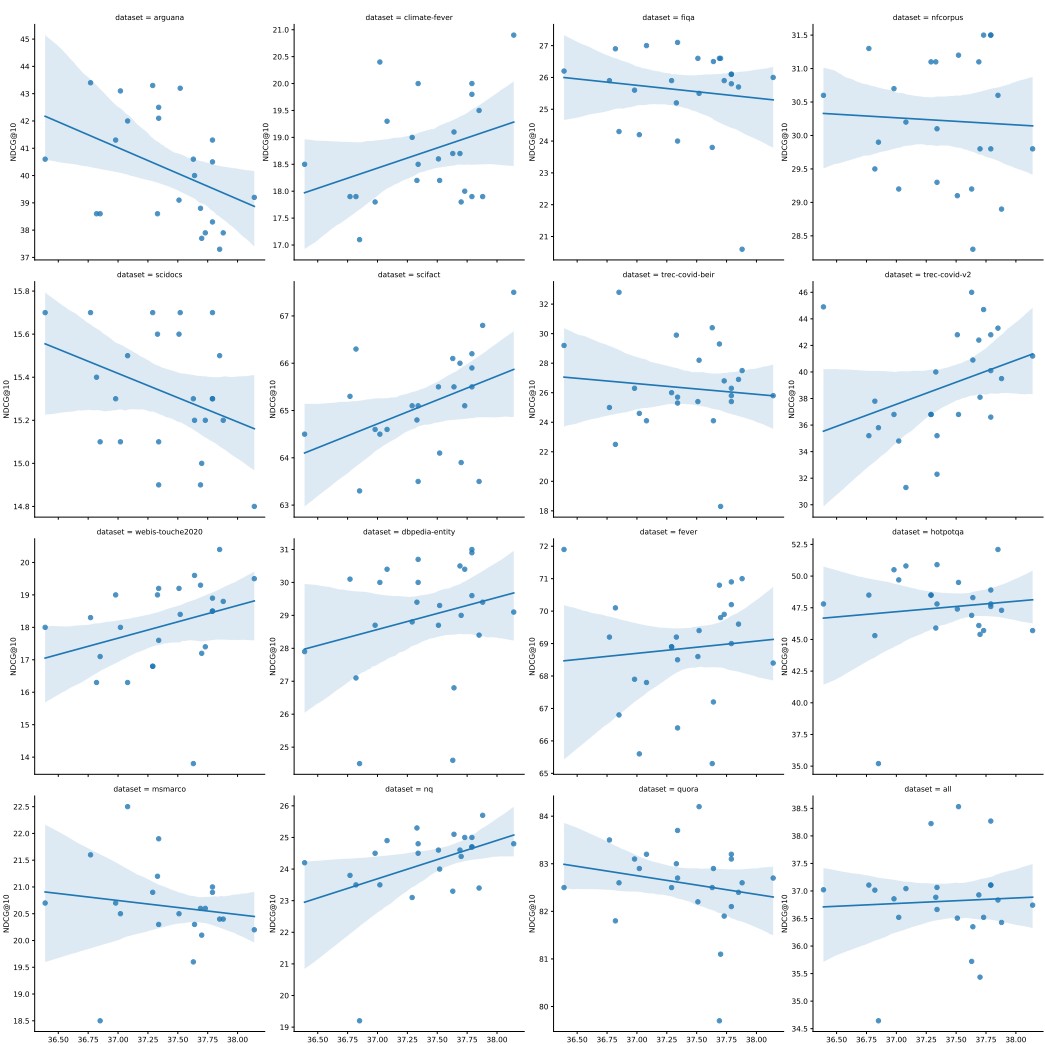

Figure 6: Full set of scatterplots of the correlation between x-axis **gender** compression (ratio of uniform to online codelength) and y-axis performance (NDCG@10), for all datasets individually, and for the average of all BEIR datasets (lower-right). Shaded region is 95% confidence interval.

of all representations in the BiasinBios dataset (272k) as well as average self-similarity of 1000 randomly sampled representations of those bigraphies, as measured by cosine similarity and by dot product. The former answers the question of how much volume the representations occupy, the latter describes the vector space via how conical (anisotropic) or spherical it is.

In Figure 8, we observe that the vector space of MultiContriever 13 is both larger volume and more obtusely anisotropic (i.e. it occupies a wider cone) than other MultiContrievers. The more obtuse anisotropy originates from MultiBert 13, as can be seen in the high variances for both seeds in cosine similarity. But the larger relative volume happens during the training of the MultiContriever and is unique to it. For MultiBert 13, L2 norm is within normal range, and the anomalous seeds are seeds 10 and 23, which both have larger norms and 5x the variance of other seeds. MultiContriever 13, however, has 1.5x the average norms of all other seeds (which have regularised and become closer in values) and 6x the variance of others. Both MultiBert 13 and MultiContriever 13 have very high variance to average cosine similarity, where the effective range of MultiContriever 13 is -0.03 to 0.53, and MultiBert 13 is 0.02 to 0.58, as compared to other models have a range of 0.28-0.32, for both types of models.

We hypothesise that this reveals a limitation of reliance on the dot product for retrieval, any operation reliant on the dot product loses information when there is a chance of a cosine similarity of zero.

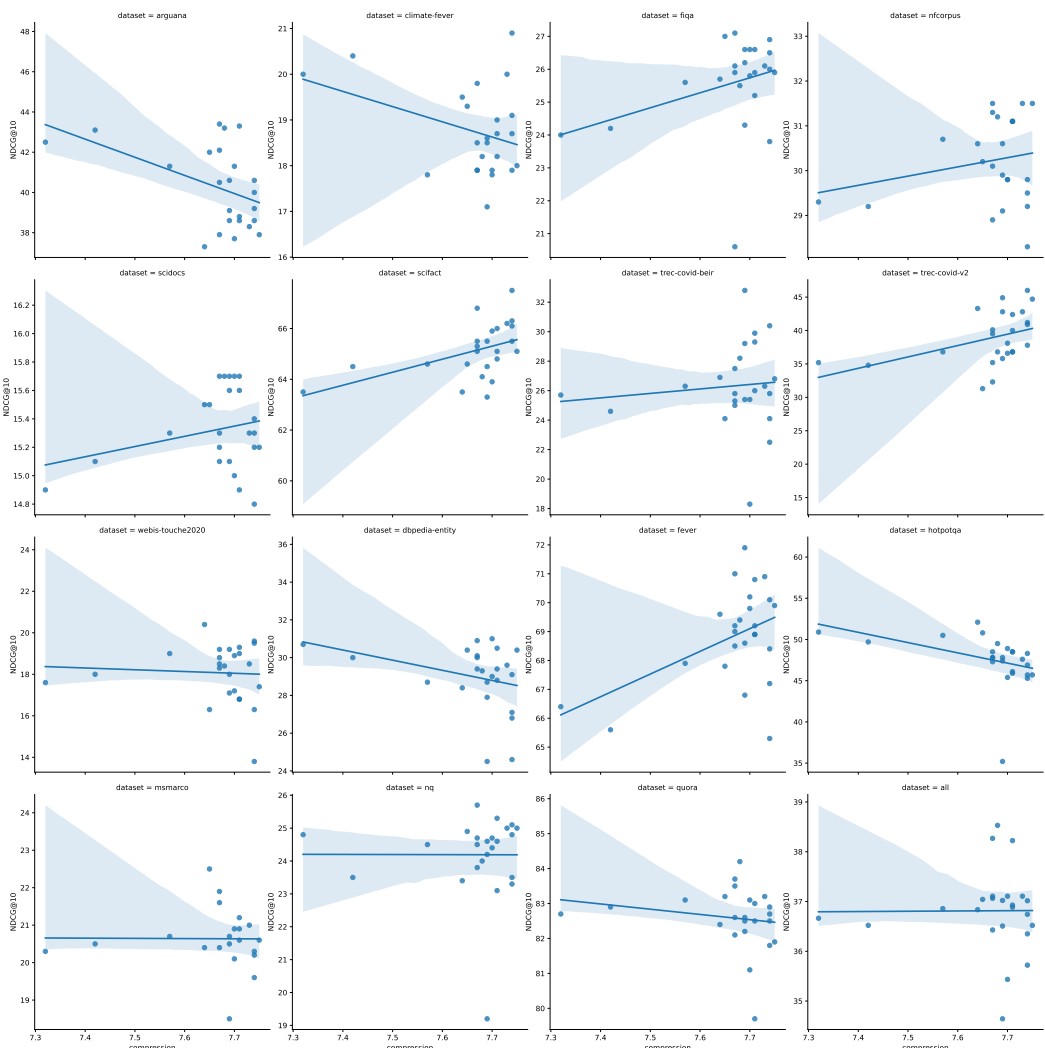

Figure 7: Full set of scatterplots of the correlation between x-axis **profession** compression (ratio of uniform to online codelength) and y-axis performance (NDCG@10), for all datasets individually, and for the average of all BEIR datasets (lower-right). Shaded region is 95% confidence interval.

We leave other investigation – such as why this would persist from a difference of only random seed initialisation, or why this issue would appear in retrieval, but not in any tasks in the MultiBerts paper, or in the contrastive training process – to future work.

We also note that seed 10 was anomalous in performance compared to the other seeds on the BEIR benchmark; not so anomalous as to be excluded, but it was reliably performing poorly. We can see the higher variance in L2 norms for 10 and 23 in MultiBerts, and then for 10 still in MultiContriever (though nothing noticeable in cosine similarity). Seeds 10 and 13 were not found to be anomalous by Sellam et al. (2022), but they did find seed 23 to display strange behaviour and be extremely unbiased (or even anti-biased) on the Winogender benchmark.

We hope that future work will use our models and continue this line of analysis.

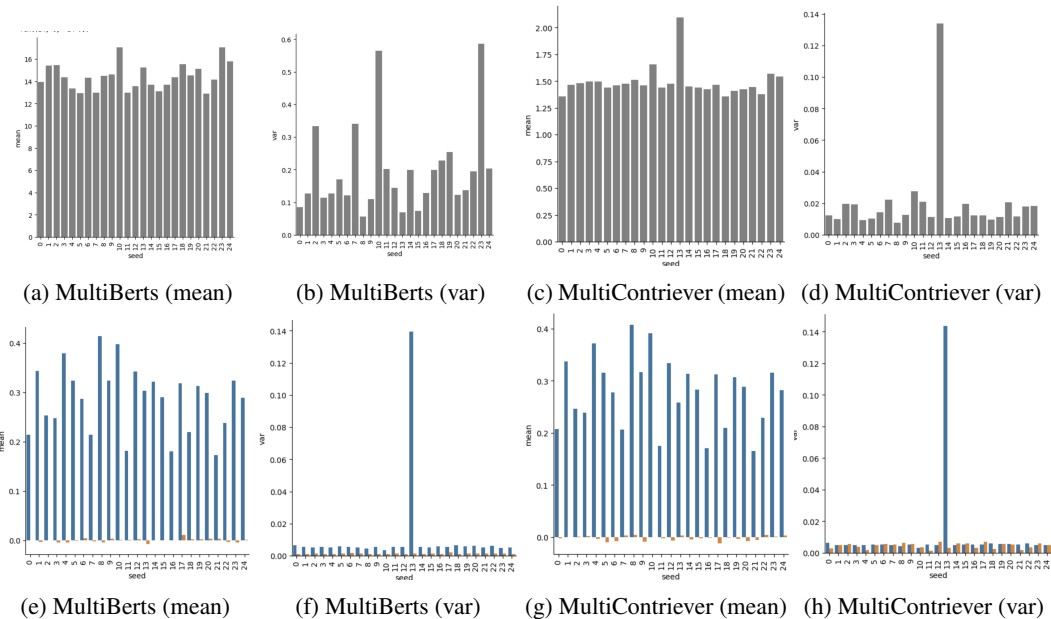

(a) MultiBerts (mean)        (b) MultiBerts (var)        (c) MultiContriever (mean)    (d) MultiContriever (var)

(e) MultiBerts (mean)        (f) MultiBerts (var)        (g) MultiContriever (mean)    (h) MultiContriever (var)

Figure 8: Top row: mean and var of L2 norms of the full BiasinBios dataset for all MultiBert and MultiContriever seeds. Bottom row: mean and var cosine similarity between 1000 random biographies.

