# OpenReview forum: "MultiContrievers: Analysis of Dense Retrieval Representations"
_ICLR.cc/2024/Conference — Submitted to ICLR 2024_

### Official Review · Reviewer_jMXz · 2023-10-30

**Soundness:** 2 fair
**Presentation:** 2 fair
**Contribution:** 2 fair
**Rating:** 3
**Confidence:** 3

**Summary:**

This paper empirically explores what information is encoded by the embeddings of dense retrievers. It leverages probing techniques to measure the degree of information that is encoded by the embeddings. It assesses the correlation between extractability of information in these embeddings and performance across 14 retrieval datasets spanning various retrieval tasks.

**Strengths:**

The exploration of embeddings' underlying information is a commendable effort. The motivation of the research is acknowledged.

**Weaknesses:**

- The encoded information of embeddings are influenced by many factors, e.g. the training data, the training algorithm. To understand the behavior of embeddings, it is necessary to take all these factors into account. Unfortunately, the paper fails to take into account any of these factors, which significantly diminishes the research significance of its conclusion.

- The paper is not quite readable. The quality of presentation is far from that of a professional research paper.

**Questions:**

Please check the posted weaknesses.

---

> ### Author Response · Authors · 2023-11-23
> **Author response: we intentionally look at contrastive training and random seed rather than training data**
>
> We respectfully disagree that within one research paper we should have included an examination of 1) all possible training data and 2) all training algorithms.  We have examined the factors of 3) random seed initialisation 4) data shuffle 5) evaluation dataset.
>
> Holding training data and training algorithm fixed was a deliberate experimental choice to control variables: by choosing the most common setups for each of those, we ensure that our work is societally relevant, and then by fixing them while we vary other experimental conditions we ensure that the work is also scientifically valid.
>
> To accept your premise that work that fixes these variables is of no utility would invalidate most of probing literature, which is a quite famous field of work influencing literally thousands of current NLP papers.

---

### Official Review · Reviewer_LvdQ · 2023-10-31

**Soundness:** 3 good
**Presentation:** 3 good
**Contribution:** 3 good
**Rating:** 6
**Confidence:** 4

**Summary:**

The authors initialize Contrievers from the MultiBERTs and evaluate on BEIR. The goal of the work is to measure how sensitive Contriever is to random initialization. There is variance on the BEIR data, but there is something not clear about whether the correct reference numbers are used, and more importantly, it remains unclear if data shuffling is more important than random initialization and BEIR may not be fully representative of supervised retrieval. In contrast, the results pertaining to gender and topic bias show low variance, but perhaps should factor in findings from previous works associated with gender bias and retrieval.

**Strengths:**

1. It will be interesting for the community to see detailed analysis pertaining to random initialization of dense retrievers. Although, it is not clear that BEIR is the correct evaluation set here, since it does not involve supervision---it could be the models show low variance when trained on supervised retrieval.

2. There is extensive experiments and analysis on BEIR and also on gender/topic bias. Although sometimes the plots are hard to read, and there are multiple concerns about the data and model choice.

3. The models and intermediate checkpoints will be released. Although it could be helpful to include at least one model that has the same weight initialization, but different data shuffling.

**Weaknesses:**

1. The evaluation tasks are designed for a specific subset of retrieval that is notoriously hard. Perhaps evaluation on fully supervised retrieval would be more informative.

2. It is interesting to study variance in retrieval, but Contriever is not especially competitive compared to more recent dense retrievers.

3. I am not sure the reference numbers for contriever are correct. I checked the contriever paper and saw ndcg@10 is 75.8 instead of 68 for Fever, and is 67.7 instead of 65 for scifact. More generally, the claim that some Contriever random seeds outperform some "supervised" models remains confusing since those models are not supervised directly on BEIR and are often outperformed by unsupervised models like BM25. Although more modern models will likely outperform both Contriever and BM25, such as GTR. Contriever paper, https://openreview.net/pdf?id=jKN1pXi7b0

4. The plots are very hard to read with blurry and small text.

5. Perhaps this is the first paper to apply probing to dense retrievers, but there are many papers that analyze gender bias of neural retrieval. These should probably be addressed. See Rekabasz and Schedl: Do Neural Ranking Models Intensify Gender Bias?

**Questions:**

Q1: After a quick search, it appears there are datasets specifically designed to measure gender bias for information retrieval. Did you consider using these datasets? How would they compare to your approach?

Q2: Is gender / topic bias only a concern for dense retrieval? What about sparse retrieval?

Q3: It does seem strange how Contriever seems to converge always to the same accuracy. Could this be due to some peculiar property of the Contriever training? Is it worth training not only on a different shuffle of the data, but perhaps removing some of the data too?

Q4: There should probably be additional clarification for "exceed the difference in per- formance from adding supervision (over unsupervised learning only)", since models evaluated on BEIR on never trained directly on BEIR. I assume the supervision is from some other retrieval data such as MSMarco.

Q5: Have you considered causes for variance in some of these datasets? For example, TREC COVID is very small (only 50 queries). HotpotQA is much larger but has one of the stranger evaluation protocols for BEIR, since the original HotpotQA relevance would typically require documents to be retrieved together and BEIR abandoned this.

Q6: Did you consider that data shuffling for Contriever training may play a bigger role than the MultiBERT seed?

Q7: "The corpus could have lower quality or less informative articles about female entities, queries about women could be structurally harder in some way." It could be helpful to clarify this with some examples.

Q8: Why did you choose contriever instead of modern alternatives? I realize that MultiBERTs allow for diversified checkpoints, but it may greatly limit our understanding when applied to retrieval. Are none of the modern retrievers compatible with BERT? Is there some option to provide multiple checkpoints for a modern retriever, perhaps through data shuffling or re-initializing some layers? Or perhaps a different retrieval approach focused on re-ranking the output of BM25 (this is only a rough idea, and I am not confident it is sufficient)?

Presentation Notes

* Figure 1 is hard to read. Perhaps remove background grid and make the reference line more prominent? Why are some plots missing the reference---should indicate if outlier is better or worse? Also the axis is so small... It could be better to include a subset here and put the rest in the appendix.

* typo: there is a large range of benchmark performance across seeds [with for] identical contrastive losses

---

> ### Author Response · Authors · 2023-11-23
>
> Thank you very much for the detailed feedback!
>
> We appreciate the recognition of the strengths of the work, and the potential value to the community of a detailed analysis of dense retrievers, as well as the resources we intend to release, and the specific analysis of gender bias.
>
> We hope we can clear up some of your questions below, and show that the choices we made were intentional and well-motivated. We also have some experiments we'd done already that should address some of your concerns, which we detail below.
>
> Overall comments:
>
> 1) We do use the correct reference numbers for contriever: the ndcg@10 performance numbers you reference are for contriever fine-tuned on MSMARCO, not for contriever fully unsupervised. The numbers we report are for the latter, as that is the accurate comparison to our work. Both of these numbers (unsupervised and supervised) are in a table in the active repo:https://github.com/facebookresearch/contriever#beir; we confirmed their accuracy by contacting the original contriever paper authors.
>
> 2) Data shuffling vs. random weight initialisation. We agree this is interesting! We have done those experiments on the effect of data shuffle, but hadn’t initially included them due to space constraints. To summarise: for that experiment, we took the best, worst, and average seed initialisation and ran them each with 5 different shuffles. We graphed the spread overall, and by dataset, and found that the shuffle did have a significant effect, though less than random seed initialisation. Two notable things from this experiment are: a) the best, middle, and worst seed initialisations mostly stayed in the same order b) the range under data shuffle differed as a property of the random seed: where the worst performing seed had a very large range but the best performing had a very small range. Based on your feedback we will add them to strengthen the findings. Note that in addition to this finding, the MultiBerts paper did extensive experiments on the influence of pretraining seed vs. other factors for the MultiBert models [see Appendix D in MultiBerts](https://openreview.net/forum?id=K0E_F0gFDgA) and found that pretraining seed mattered most, so this is well supported.
>
> 3) Other gender bias work: [Rekabsaz and Schedl (2020)](https://arxiv.org/pdf/2005.00372.pdf) define bias as the genderedness of retrieved documents based on lexical terms, making the implicit normative statement that lack of bias means equal representation of male and female documents in non-gendered queries: essentially an independence assertion (as defined by [Barocas, Hardt, and Narayan](https://fairmlbook.org/classification.html)). Variants of this approach are taken by the follow on work surveying gender bias metrics [Klasnja et al (2022)](https://fairmlbook.org/classification.html). This is quite different to our approach, which looks at performance disparity between queries that require male and female gender information to answer. Our approach has more immediate practical utility for a real world retriever, and also ties in to the work on information theory by restricting to queries that require gender information. The aforementioned works release subsets of MSMARCO, which we did examine and use in initial tests early in this work, but decided that they were unsuitable because of the very different metrics and approaches. We agree with you that the paper would be stronger if we add this context, so we will include it in the final version!
>
> 4) Supervised vs. unsupervised retrieval: we had two motivations in using unsupervised retrieval. First. unsupervised retrievers are in wide usage and thus examining them is extremely societally relevant. Second, unsupervised retrieval has cleaner experimental conditions. Supervision adds an additional experimental variable: the fine tuning dataset and its data shuffle, which is unwieldy and costly in an already extremely compute intensive investigation (necessarily so to get many random seeds). Supervision also muddies the analysis: if we’re looking at generalisation capabilities from the objective to the retrieval dataset (which we are) then if we fine-tune we have to have some way of measuring similarity of the dataset we fine tune on to the dataset we test on, and methods for that aren’t very well established. In practice, when you know the domain, you fine tune, but you don’t always have that luxury (which is why unsupervised retrievers are so common). And even when people fine tune, they often contrastively train first, so these results are of value to the whole community and can provide a strong base for further research into the fine tuning domain.

---

> > ### Author Response · Authors · 2023-11-23
> >
> > 5. Use of Contriever: we chose contriever largely because of societal relevance. Contriever is contemporary with GTR, and has two orders of magnitude more monthly downloads (see [HF unsupervised contriever repository](https://huggingface.co/facebook/contriever), vs. [GTR large](https://huggingface.co/sentence-transformers/gtr-t5-large) and [base](https://huggingface.co/sentence-transformers/gtr-t5-base) repos). As you anticipate, we also chose it because of the opportunity to build on top of MultiBerts for breadth of experiments, as it is rare to have the ability to study across so many random initialisations. We consider it most valuable to study this controlled setting that also mirrors real world usage, and less valuable to analyse the very latest and fanciest retriever.
> >
> > 6. We very much apologise for the blurriness of the plots! We will rearrange to make the plots in Figure 1 bigger, and we realise that some figures accidentally got included as pngs instead of pdfs. We have remedied this, it is a very easy fix.
> >
> > Questions:
> >
> > Q1: See #3 above for why we did not use the existing datasets on gender bias. This will all go into a discussion in the methods in the final paper.
> >
> > Q2: We scoped this paper to examine the information that dense retrievers encode, and how strongly they encode it. Information theoretic compressibility also makes most sense as an investigation of dense retrievers, but it could be expanded to sparse and the comparison would be very interesting, though isn’t within the scope of this work, which can’t expand to be comprehensive about all possible sources of gender bias. We’ll add a note for future work!
> >
> > Q3: This is quite commonly the case with generalisation tasks: the pretraining task is easier, so models with different capabilities can hit the same accuracy. [McCoy et al 2020](https://arxiv.org/abs/1911.02969) found the same thing with their results: near identical accuracy on fine-tuning sets and wildly different accuracy on later challenge sets. We will add this explanation to the text to clarify.
> >
> > Q4: You are correct, supervision is on MSMARCO for the comparisons. We will amend this.
> >
> > Q5: Yes absolutely the datasets are both vastly different domains and different sizes! However, the variance is still relatively common across datasets, very large datasets like Fever and NQ have similar ranges to very small ones like Arguana. This knowledge played into our finding that there were no significant correlations: some of the small datasets did seem to have correlations (arguana and trec-covid) but both were too small for us to draw conclusions. As an additional note though: we test on two versions of trec-covid, one that was created by the BEIR authors to augment the original dataset, so we consider the second one, the “beir” version, more reliable (both very small in queries, just differences in number of annotations in the retrieval corpus). Apart from that, we don’t make any dataset specific conclusions, we simply plot them all separately as otherwise the aggregation masks significant variability even in the big datasets. We will add all this discussion to the paper, it’s a good point! We’ll also add a statistical significance measure which will incorporate the size.
> >
> > Q6: See #2 above for dataset shuffle.
> >
> > Q7: Absolutely, we will add some examples. One example from research comes from [Sun and Peng (2021)](https://aclanthology.org/2021.acl-short.45.pdf), which finds that wikipedia articles tend to be written very differently for male and female entities, with the male ones being better structured into separate sections for career and personal life whereas female ones are more interleaved. This would likely make it easier to obtain high relevance scores for male pages for a retriever. This is just one example, we will add a few more.
> >
> > Q8: See #5 above for most of this. In regards to BM25 + reranking, we scoped this work to focus on dense retrieval since more of the existing work incorporated rerankers, and we were more interested in the information theoretic effects on dense retrievers. This would be good future work though, and we’ll add a note. We think the interaction of our findings with lexicalisation would be very informative for someone to do with the resources we release.
> >
> > Final two small notes:
> >
> > We’ve corrected the typo (thanks!) and will make all the presentational improvements you suggest right away!
> >
> > Figure 1 has only one plot missing a reference (the original trec-covid-v2): the original contriever wasn’t tested on this so it was missing from our reference data. We will run the original contriever on it and add the reference line.

---

### Official Review · Reviewer_wTy7 · 2023-10-31

**Soundness:** 2 fair
**Presentation:** 2 fair
**Contribution:** 3 good
**Rating:** 3
**Confidence:** 3

**Summary:**

This paper presents an information theoretic analysis on the dense retrieval representation, obtained using the contrastive loss, by providing the extractability metric. Focusing on two types of information – topic and gender, the paper observes 1) the contrastive learning increases the extractability of both topic and gender, 2) but the extractability is poorly correlated with the IR benchmark performance, and 3) the allocational gender bias is observed, but the bias is not reduced after removing the relevant gender information (using the null space projection). Overall, this work presents a series of interesting observation on information bias entailed from the dense retrieval representation.

**Strengths:**

- The proposed probing analysis based on the extractability for exploring the dense retrieval representation is quite interesting and novel.
- The reported analysis on topic and gender is also valuable and useful; The extractability does not necessarily entail the retrieval performance, and the bias such as gender bias is not originated from the representation itself, etc.

**Weaknesses:**

- The probing analysis such as the correlation b/w the extractability and the retrieval is explored well. But, it is unclear how to applying the current probing analysis to obtain better retrieval or application tasks. How the retrieval method is modified such that the extractability is helpful to improve the performance?
- The current experiment is restricted to only two types of information – topic and gender. An extension to other types of bias is desirable.
- The extractability is considered as the only metric for this probing method, but other metrics need to be discussed and considered for extracting target types of information.

**Questions:**

- How the probing model is designed?
- It seems that the retrieval performance is reported for all queries. What is retrieval performance when using the topic/gender-related queries are examined?
- What is the motivation that the "contrastive learning" to obtain the dense representation can further enhance the extractability?

---

> ### Author Response · Authors · 2023-11-23
>
> Thank you for recognising the novelty of our contributions and the value to the community of our analysis.
>
> We believe some of the perceived weaknesses to potentially be confusion about the goals of our paper (though the summary was accurate), so we hope the below clarifies!
>
> 1) Extractability is the only metric for this probing method because the metric *is* a property of the method: MDL probing gives you the metric of extractability. Probing methods tend to use either accuracy or extractability, and extractability has been quite strongly shown to be much more reliable than accuracy (see Voita and Titov 2020, who developed MDL probes, for an analysis, as well as Hewitt and Liang 2019 on probe design). Thus there is no utility to using accuracy. Extractability is also the only metric that has been shown to be related to generalisation (Lovering et al 2021) and to fairness (Orgad et al 2022). So the one metric is sufficient to make our point. The important things to vary for probing are the *dataset* used; this ensures it is not an accidental property of the dataset. For this reason we used both BiasinBios and Wikipedia (in 9 different random splits) to verify our findings.
>
> 2) We focused on only topic and gender because that was what we could find available dataset labels for, for things that were likely to matter for retrieval. We’re more than happy to extend this work to additional labels if more suitable datasets are made. We’ve also enabled future work to do this, as we are releasing all the models and the code upon publication. The compute and time intensive parts of this work are the model training and the experimental code: running probing on a dataset can be done with one of our shell scripts and easily reported to wandb automatically. We will add a note to this effect to suggest the future work.
>
> 3) The focus of the paper is explicitly scoped to be analysis, so the goal is not to improve performance. There are however actionable takeaways from our findings! We included them in the conclusion, but we will edit it to make sure they are highlighted more clearly. They are: 1) train over many random seeds *before* doing other additional work to improve performance, as this is a strong source of variability 2) the shape of the vector space is an early indicator of how good the contriever will be and 3) gender bias in retrieval needs to be corrected by looking at the corpus and the queries, not just the representations. These are all very actionable and will improve both performance and bias.
>
> Questions:
>
> Q1: The probing model is exactly as designed in Voita and Titov (2020): https://aclanthology.org/2020.emnlp-main.14.pdf.
> There is also an excellent blog post on the topic (Voita usually does blog posts): https://lena-voita.github.io/posts/mdl_probes.html.
> To summarise here, an MDL probe measures online codelength, which quantifies the compressibility of the data. To do that, the data is split into many small sections (1…N) and a classifier is trained to predict labels Y from representations X. Then the classifier is trained again on section 1 + section 2, etc to N. If Y is very extractable from X, it will also be very compressible, and each additional split will not be very hard to transmit because the regularity will have been learnt with little data. If Y is not extractable from X (as in random embeddings for instance) then there will be no compression at all and the effort to transmit will equal the amount of data. The benefit of this method is that it is not sensitive to the parameters of the classifier (which using accuracy is, you can just overparameterise the classifier and memorise noise).
>
> Q2: Yes we do that experiment! Please see Figure 3, where a) and b) are all queries and c) is only gendered queries, as you suggest (and d is control). We also look at gendered vs. non gendered queries in Figure 4.
>
> Q3: The motivation for contrastive learning is to leverage large amounts of unstructured data to learn good representations for retrieval: supervised retrieval datasets are extremely expensive to make (annotators have to select the relevant documents from a corpus of potentially millions of documents) so it is extremely common to take a contrastive approach instead. This is what contriever does. There are many tricks to this process, which are described in detail in the contriever paper: https://openreview.net/forum?id=jKN1pXi7b0. For our purposes, the important thing is that it optimises a contrastive loss, rather than an MLM objective, which should encourage different properties in representations. That difference is part of what we are analysing, and that analysis is part of our contribution.

---

### Official Review · Reviewer_rp3a · 2023-11-02

**Soundness:** 2 fair
**Presentation:** 2 fair
**Contribution:** 3 good
**Rating:** 6
**Confidence:** 4

**Summary:**

The paper analyzes a suite of Contriever retrieval models, each initialized from a different pre-trained BERT model. The primary aim of the paper seems to be examining the extent to which contriver representations capture aspects such as gender and occupation (referred to as topic in the paper) of the subject of a document and to what extent this information correlates with retrieval benchmark performance. The paper presents results showing: 1) contriever models see a large variation in performance based on random seed 2) while gender and topic are more extractable from contriver representations than BERT, the ratio of the two is more extractable from BERT. 3) Extractability of gender and topic dont correlate with performance on the retrieval benchmarks.

**Strengths:**

- The paper presents an analysis of retrieval model representations that has not been done before.
- The results presented may be useful for future model development work for retrieval tasks.

**Weaknesses:**

- Motivation and framing: Examining the extractability of gender and occupation and attempting to correlate this with benchmark performance seems undermotivated and distracting - it's unclear why one would expect these two pieces of information to be crucially important for performance on the datasets. (on the other hand, the analysis with gendered queries seems reasonable)
- Experimental details: The experimental section seems less than ideal in rigor, the writing in the paper is a bit scattered, and the figures are quite poorly labeled for a paper that is largely result-driven.

**Questions:**

- I would highly encourage referring to "topic" in the paper as "occupation" - this is much clearer and seems much more true to the data used for analysis (even the original paper seems to refer to this as occupation: https://arxiv.org/abs/1901.09451). The current presentation also makes it seem like the papers scope is much larger than it actually is. Else, please rationalize the scope and the framing of the analysis in greater detail.
- Several claims in the paper ("This highlights the gap between ... a limitation of the benchmark", "lack of correlation between extractability and performance points to a mismatch between the self-supervised contrastive training objective ...") seem to imply that there is a problem with the BEIR benchmark tasks or the training objective for contriver because while the training improves extractability of gender and occupation it does not correlate with the benchmark performance. This is not a reasonable inference - an alternative is that the two attributes examined are insufficient for explaining the underlying signals captured by the benchmark tasks and for studying the influence of training. For example training may be improving the extractability of several other attributes not examined in this paper which may better explain benchmark performance. Please soften these claims and discuss the limitations of the attributes examined in this paper (which would of course not be a slight on this work).
- Table 1 and Figure 1: Thank you for this analysis, this is interesting. While I understand that the analysis is somewhat tangential to the point of the paper, I would recommend some changes to strengthen this analysis: 1) It seems a bit misleading to report the max-min difference for various seeds as a measure of variance based on the seed. Please also report the standard deviation in performance across seeds - this is more common for reporting this kind of variance: https://arxiv.org/abs/2302.07778. 2) Please report similar statistics for other metrics like Recall@100. It may be the case that the large gaps in performance are a result of the metric choice (eg exponential gain in computation of NDCG) and the presence of unjudged documents in retrieval datasets. Deeper rank metrics will alleviate the latter issue. It would also be illustrative to understand the Recall since dense retrieval systems are usually used for a high recall first stage retrieval. 3) A few of the datasets used in the BEIR benchmark are quite small (~50 queries). Please consider computing statistical significances for the max-min differences. The differences are unlikely to be significant for all the datasets. 4) It is unclear why the seeds influence performance so much - please consider discussing this in greater detail - do the representations change massively due to random seeds? Do the score distributions for positive vs negative change across seeds? (similar to: https://proceedings.mlr.press/v162/menon22a.html) etc.
- Please add a main number/caption for all figures and label the axis of plots aptly - they are missing in many places.
- It's unclear what the procedure for measuring extractability of the gender:topic ratio is - please clarify this better.
- Please consider citing: https://dl.acm.org/doi/10.1145/3234944.3234959

---

> ### Author Response · Authors · 2023-11-23
>
> Thank you very much for the extremely thoughtful review! We appreciate your interest in the paper and your detailed notes on it very much.
>
> We also appreciate your recognition of the novelty and utility to future work in this area, and your interest in our findings on random seed variability.
>
> We’ve made most of the revisions you suggest, and detailed some experiments that we already have that address a few of your concerns. We discuss everything below.
>
> Overall comments:
>
> We apologise for the figures being sometimes subpar in labelling and resolution, we made a couple of mistakes using pngs instead of pdfs, and we see the missing labels also. All of these are trivially fixable to improve presentation, and we will do so for the final. We’ll also do a few more editing passes for language and clarity.
>
> On correlation experiments: the motivation for looking at the correlation was twofold: first, since contriever training so strongly increased extractability, there was an interesting question as to fit between the proxy task in contriever training and the retrieval benchmark: one might reasonably expect, if the proxy task increases it, it to also be important for performance (or at least correlated with, due to some shared underlying factor), but that was an unknown. Second, it came up as a desired addition when we sent a very early draft for feedback to some authors of relevant work. So we added it at that stage, which is perhaps why it doesn’t seem to fit perfectly. We do still find it to be a valuable finding (and were asked for it) so we’ll do a few more drafts to make sure that it’s less distracting, and reorganise to focus on it a little bit less.
>
> Questions/Suggestions:
>
> Q1: Very fair, we will change it, you’re right that it’s overscoped. It is an artefact of when we had broader experiments on some of the other gender datasets (which included other topic-like metadata) which we then found to be too low quality and had to remove (footnote 3 describes the removal).
>
> Q2: Also a very good suggestion, we will clarify and soften those claims, and discuss those limitations, we agree with your revision fully.
>
> Q3: Regarding Figure 1:
>
> 1) We visualised variability via histograms of performance for comparability to the MultiBerts paper, since that is what they used. For the final, we’ll also add standard deviation for comparability to other work in this domain (like in that reference, which is excellent), thank you for the suggestion.
>
> 2) Very good point: we did analyse a variety of metrics in our experiments, and it doesn’t change the results: we looked at whether Hole@ metrics seemed to be at play and weren’t able to find anything there, and Recall@100 tracked largely the same as NDCG@10 or 100, including the best and worst seeds (though there is a little bit of shuffling around with the seeds in the middle). In the final, we’ll include our graphs and stats for Recall@100 for all experiments. Your point about frequent usage as a high recall first filter is a good one.
>
> 3) We will include stat sig for all datasets in the final: since we saw the variance in all datasets (even the very large ones), we knew it not to be a property of size, but it’s still a good addition and we will add it.
>
> 4) Our analysis of the vector spaces of seed 13 (Appendix D) did illuminate some other differences in other random seeds, but predominantly the weakness of seed 10. There weren’t other major differences in anisotropy or vector space volume across other models. Are the similarity measures you’re thinking of something like Linear-CKA? We agree that would be interesting and can add for the final. We had not considered doing the approach of looking at the positive vs. negative distributions, thank you for that idea, it’s very cool. That may be a bit harder to get before the final, but we’ll look into it! And certainly suggest it as follow on work if we can’t, since we’re realising everything.
>
> Q4: We will improve all the figures, thank you! (see overall comment also).
>
> Q5: We will clarify the ratio of gender:topic extractability procedure in the text, our wording was just a bit unclear. This is the ratio of extractability of gender to extractability of topic (not a separate thing measured). We do this since it isn’t only the absolute number that matters for model shortcutting, but also the relative extractability of various information that a model could use. So Figure 2c actually contains the information of 2a and 2b, just displayed differently so that the change in the ratio can be seen. We’ll make this clearer.
>
> Q6:  Thank you, we will cite that paper!

---

### Author Response · Authors · 2023-11-23
**Overall comment**

We thank all the reviewers for their feedback, and for noting that our work does novel experiments into analysis of dense retriever representations, and that the results and the resource are of actionable value to the community.

We apologise that the plots were a bit too fuzzy in axis labels due to a file type error of including some pngs instead of pdfs, we have remedied this and made all other presentational improvements that were recommended.

---

### Meta-Review · Area_Chair_4hQG · 2023-12-05

**Metareview:**

This is the first paper to present analyses that examine the information encoded in dense retriever representations. The authors utilise probing techniques and analyse a number of Contriever retrieval models. They examine the extent to which their representations capture aspects of gender and topic, as well as the extent to which this information correlates with retrieval benchmark performance.
While the paper presents quality work, the reviewers have raised a number of concerns (we note, however, that Reviewer jMXz was not taken into account due to the very short review and little feedback to the authors). For reviewer wTy7, it seems the authors have addressed their concerns. For the other two reviewers, we note that there seem to be a number of points where the authors agree with the reviewers' concerns (e.g., scope not clear, claims not fully substantiated, seed influence, to name a few) and promise to implement the suggestions and add new results; however, these are not actually presented in the rebuttal, which makes it difficult to fully appreciate the significance of paper in its current form. Furthermore, I would agree with one of the reviewers that “this work is highly limited to a specific class of models and not representative of retrieval behaviour at large”. While that reviewer thinks "the work will be valuable and perhaps inspire future work to conduct a more thorough study”, I would still think that presenting a thorough study would be the minimum for acceptance in such a venue as ICLR. A final point regarding the reasoning behind the motivation of certain experiments: the fact that these experiments “came up as a desired addition when we sent a very early draft for feedback to some authors of relevant work” is not a good enough reason.

**Justification For Why Not Higher Score:**

This work is highly limited to a specific class of models and not representative of retrieval behaviour at large.

**Justification For Why Not Lower Score:**

NA

---

### Decision · Program_Chairs · 2024-01-16

Reject